# Outcomes of Community-Based Systematic Screening of Household Contacts of Patients with Multidrug-Resistant Tuberculosis in Myanmar

**DOI:** 10.3390/tropicalmed5010002

**Published:** 2019-12-25

**Authors:** Nang Thu Thu Kyaw, Aung Sithu, Srinath Satyanarayana, Ajay M. V. Kumar, Saw Thein, Aye Myat Thi, Pyae Phyo Wai, Yan Naing Lin, Khine Wut Yee Kyaw, Moe Myint Theingi Tun, Myo Minn Oo, Si Thu Aung, Anthony D. Harries

**Affiliations:** 1Center for Operational Research, International Union Against Tuberculosis and Lung Disease, Myanmar Office, Mandalay 05021, Myanmar; aungsithu@theunion.org (A.S.); drayemyatthi@gmail.com (A.M.T.); pyaephyowai@theunion.org (P.P.W.); nainglinn.zhaohong.yan@gmail.com (Y.N.L.); dr.khinewutyeekyaw2015@gmail.com (K.W.Y.K.); dr.moemyint86@gmail.com (M.M.T.T.); dr.myominnoo@gmail.com (M.M.O.); 2Center for Operational Research, International Union Against Tuberculosis and Lung Disease, South-East Asia Office, New Delhi 110016, India; SSrinath@theunion.org (S.S.); AKumar@theunion.org (A.M.V.K.); 3Center for Operational Research, International Union Against Tuberculosis and Lung Disease, 75006 Paris, France; adharries@theunion.org; 4Yenepoya Medical College, Yenepoya (Deemed to be University), Mangaluru 575018, India; 5National Tuberculosis Programme, Department of Public Health, Nay Pyi Taw 15011, Myanmar; dr.sawthein2010@gmail.com (S.T.); sta.ntp@gmail.com (S.T.A.); 6Department of Infectious and Tropical Diseases, London School of Hygiene and Tropical Medicine, London WC1E 7HT, UK

**Keywords:** multidrug-resistant tuberculosis, household contact, screening, TB diagnosis, yield, operations research

## Abstract

Screening of household contacts of patients with multidrug-resistant tuberculosis (MDR-TB) is a crucial active TB case-finding intervention. Before 2016, this intervention had not been implemented in Myanmar, a country with a high MDR-TB burden. In 2016, a community-based screening of household contacts of MDR-TB patients using a systematic TB-screening algorithm (symptom screening and chest radiography followed by sputum smear microscopy and Xpert-MTB/RIF assays) was implemented in 33 townships in Myanmar. We assessed the implementation of this intervention, how well the screening algorithm was followed, and the yield of active TB. Data collected between April 2016 and March 2017 were analyzed using logistic and log-binomial regression. Of 620 household contacts of 210 MDR-TB patients enrolled for screening, 620 (100%) underwent TB symptom screening and 505 (81%) underwent chest radiography. Of 240 (39%) symptomatic household contacts, 71 (30%) were not further screened according to the algorithm. Children aged <15 years were less likely to follow the algorithm. Twenty-four contacts were diagnosed with active TB, including two rifampicin- resistant cases (yield of active TB = 3.9%, 95% CI: 2.3%–6.5%). The highest yield was found among children aged <5 years (10.0%, 95% CI: 3.6%–24.7%). Household contact screening should be strengthened, continued, and scaled up for all MDR-TB patients in Myanmar.

## 1. Introduction

Myanmar is one of the 30 high tuberculosis (TB) and multidrug-resistant TB (MDR-TB) burden countries in the world. In 2017, of the estimated 14,000 MDR-TB cases in Myanmar, 3281 were diagnosed and 2666 were enrolled for treatment, indicating a significant gap in case detection and treatment [1]. Similarly, of the estimated 191,000 TB cases, only 132,025 were notified and treated. To reduce the TB and MDR-TB burden, it is essential to diagnose TB and MDR-TB early and provide quality assured treatment [2,3]. Early diagnosis and treatment reduce morbidity, mortality, and transmission of TB and MDR-TB in the community. 

Close contacts of active TB and MDR-TB patients are at high risk of TB infection and disease. A systematic review reported a pooled yield of 3.4% active TB among close contacts of active TB, with the incidence being highest during the first year of exposure [4]. Another systematic review reported that the prevalence of active TB among household contacts of drug-resistant TB was as high as 7.8% [5]. Hence, there is a strong recommendation to screen all household contacts of MDR-TB patients for active TB, and if they are diagnosed with active TB, to initiate them on treatment as soon as possible [6]. 

The International Union against TB and Lung Disease (The Union) has been implementing a community-based MDR-TB care (CBMDR-TBC) project in Myanmar to support the National Tuberculosis Programme’s (NTP) programmatic management of DR-TB since 2015. Due to the high presumed prevalence of TB among household contacts of MDR-TB patients, a systematic screening algorithm including a combination of screening methods (symptoms and chest radiography) and diagnostic tests (sputum smear microscopy and Xpert MTB/RIF assay) to screen for active TB and MDR-TB was incorporated as a key component of the CBMDR-TBC project. In early 2016, community volunteers and focal nurses were trained under the CBMDR-TBC project to implement this screening for all household contacts of index MDR-TB patients in project townships, and the project started systematic data collection (which included dedicated recording and reporting systems) of this activity in March 2016. 

To date, there has been no published report from Myanmar describing the process of screening household contacts of MDR-TB patients, how well the screening algorithm was followed, and the yield of active TB among those screened. Therefore, in this study we assessed: (a) the proportion of household contacts who were screened for active TB using the systematic screening algorithm (the proportion who were screened using symptoms and chest radiography and those with TB symptoms who were investigated for active TB using sputum smear microscopy and/or Xpert MTB/RIF assay), (b) the socio-demographic characteristics associated with screening of TB according to the algorithm, (c) the yield of active TB, and (d) socio-demographic characteristics associated with the diagnosis of active TB during one year of the implementation of the project. 

## 2. Materials and Methods

### 2.1. Study Design

This was an analysis of routinely collected program data. 

### 2.2. Setting

#### 2.2.1. Country Setting

Myanmar is a lower middle-income country with a population of 51 million. Geographically, the country is divided into 15 states/regions, which are further administratively divided into 412 townships. The country bears a high burden of MDR-TB along with its neighboring countries such as China, India, Bangladesh, and Thailand. The Programmatic Management of DR-TB was initiated in 2011 as part of the NTP’s National Strategic Plan (2011–2015) to control TB in Myanmar [7]. Systematic contact tracing for all household contacts of MDR-TB patients is one of the key activities of the NTP’s National Strategic Plan (2016–2020) [8]. The NTP recommends active screening of all household contacts of index MDR-TB patients and upfront use of Xpert MTB/RIF for investigating those with presumptive TB [9]. In order to facilitate Xpert MTB/RIF testing, the NTP has rolled out Xpert MTB/RIF machines in Myanmar since 2012, and by 2016 there were 65 Xpert MTB/RIF functional machines in the country.

#### 2.2.2. Project Description and the Implementation of Systematic Screening

The CBMDR-TBC project was started in 2015 in collaboration with The Union and NTP to support the treatment initiation and adherence among MDR-TB patients in 33 townships across four states/regions in the upper part of Myanmar. Details of the CBMDR-TBC project have been described elsewhere [10]. Briefly, under the project, each township has a focal nurse who visits the index MDR-TB patients’ house monthly to monitor treatment adherence and side effects and provide health education and psychosocial support. The project also assigns a community volunteer for each patient who visits the patient’s house daily in the evening to provide directly observed treatment. Focal nurses supervise the volunteers, and the project managers of the CBMDR-TBC project in turn supervise the focal nurses. 

In 2016, a systematic screening of the household contacts of MDR-TB patients was incorporated into the project. The focal nurses and community volunteers were trained to implement the TB screening for all household contacts of index MDR-TB patients and the systematic data collection. A household contact is defined as “a person who shares the same enclosed living space for one or more nights or for frequent or extended periods during the day with the index case during the treatment or during the three months before the commencement of the current treatment”. 

The trained focal nurse facilitates the screening of household contacts for TB once the MDR-TB patients are diagnosed and every six months thereafter using a screening and investigation algorithm as described in Figure 1. During the home visit to the MDR-TB patients, the focal nurse screens each household member using a symptom-based questionnaire and refers these members for a chest radiograph. Those with symptoms and/or abnormal chest radiograph submit one early morning sputum sample for Xpert MTB/RIF and two samples, one spot and one early morning sputum, for smear microscopy for acid-fast bacilli (AFB). The focal nurses are trained to instruct contacts on how to expectorate sputum according to the guidelines. The instruction includes: First, rinse the mouth with clean water; second, take a deep breath in and out for three times; third, take one deep breath and cough forcefully; and finally, spit the sputum into the sputum container provided. Those with no symptoms are closely monitored unless the chest radiograph is abnormal, at which point the patient is referred to a TB specialist for further assessment. The contacts who are positive on smear microscopy for AFB and/or Xpert MTB/RIF assay are diagnosed with active TB. Those with negative results on smear microscopy or the Xpert MTB/RIF assay but have an abnormal chest radiograph are referred to the TB specialist for clinical evaluation and a decision on whether there is a clinical diagnosis of active TB. Each household contact is line listed and given a unique contact registration number. The nurse records the results of this screening process in the MDR-TB household contact screening register (Appendix A).

The focal nurses are responsible for screening the household contacts of MDR-TB patients as soon as the index MDR-TB patients are diagnosed. However, not all household contacts of index MDR-TB patients are screened immediately for several reasons, and therefore there could be a considerable delay between the diagnosis of an index case and the initial screening of contacts. In addition to 6-monthly systematic screening, the volunteers and nurses also check whether any TB-related symptoms have developed in household contacts during their regular home visits. If a household contact reports any TB-related symptom before the scheduled screening appointment, the focal nurse facilitates the evaluation of such patients for active TB in line with the algorithm. In addition, the focal nurses and volunteers provide TB health education and support for household infection control measures. They receive periodic training on systematic screening and investigation of active TB in household contacts conducted by the project managers using a standardized training package. The training includes steps to identify household contacts, the conduct of symptom screening, the referral of persons for diagnostic investigations for active TB and drug-resistant TB, follow up, linkage to treatment if required, education, and support for infection control measures.

Every month, the information from the MDR-TB household contact screening register is entered into an electronic database (developed using EpiInfo version 7.2 software) by the project’s data entry operators, and the data are checked and validated by the monitoring and evaluation officer of the project.

### 2.3. Study Sites and Population

The study population includes all household contacts of index MDR-TB patients enrolled for TB screening in 33 townships of the CBMDR-TBC project in Upper Myanmar between April 2016 and March 2017. The index MDR-TB patients of the contacts included in this study were MDR-TB patients who were newly initiated on treatment between April 2016 and March 2017 as well as those who were already on treatment before April 2016.

### 2.4. Sources of Data, Data Variables, and Data Collection

We used secondary data routinely collected in the electronic database. Data variables of household contacts of index MDR-TB patients included: Contact registration number; registration date for contact screening; age; sex; history of previous TB; HIV status; history of diabetes mellitus; clinical information on symptoms such as cough, fever, weight loss, hemoptysis, lymph node enlargement, and night sweats; and results of diagnostic investigations such as sputum smear microscopy, Xpert MTB/RIF assay, and chest radiography and the treatment registration number of their index case. These data were extracted from the electronic database. 

### 2.5. Analysis and Statistics

The demographic and clinical characteristics of the household contacts were described using numbers (proportions) and medians (interquartile ranges). We assessed the proportion of the household contacts with TB symptoms and of those, the proportion who underwent further sputum evaluation (smear AFB and/or Xpert test MTB/RIF). We used binomial logit models to study the association between measured demographic and clinical characteristics and the odds of further sputum evaluation according to the screening algorithm. 

The yield/proportion of TB was calculated by dividing the number of TB or MDR-TB cases diagnosed by the number of household contacts screened for TB. We also calculated the yield of active TB across various measured demographic and clinical characteristics. The prevalence ratios of active TB across various measured demographic and clinical characteristics were estimated using binomial log models. STATA software (version 12.1, copyright 1985–2011 StataCorp LP, College Station, TX, USA) was used for all analysis. The 95% confidence intervals (CIs) for proportions, odds ratios, and prevalence ratios were adjusted for clustering at the township and household level using cluster robust standard error estimates. 

### 2.6. Ethics

Ethics approval was received from the Myanmar Ethics Review Committee, Department of Medical Research, Ministry of Health and Sports, Myanmar (Approval number: Ethics/DMR/2017/084) and the Ethics Advisory Group of International Union Against Tuberculosis and Lung Disease, Paris, France (EAG number: 120/16). Permission to conduct the study was granted from the National Tuberculosis Programme, Ministry of Health and Sports, Myanmar.

## 3. Results

There were 620 household contacts of 210 index MDR-TB patients who were enrolled for systematic screening for active TB. Of those enrolled, all were screened for symptoms and 505 (81%) also underwent chest radiography. There were 240 (39%) contacts who had one or more TB symptoms and were eligible for sputum smear microscopy and Xpert MTB/RIF testing. Of those eligible, 169 (70%) underwent sputum smear microscopy and/or an Xpert MTB/RIF assay. The remaining 71 (30%) contacts did not undergo either of these tests, though some were evaluated clinically by the TB specialist (Figure 2). As a result of all these investigations and clinical evaluations, 24 contacts (3.9%, 95% CI: 2.3%–6.5%) were diagnosed with active TB (seven were bacteriologically confirmed, including two with Rifampicin-resistant TB, and 17 were clinically diagnosed). The number of household contacts screened to diagnose one case of active TB was 26 (95% CI: 15–44). 

### 3.1. Characteristics Associated with Following the Systematic Screening Algorithm among Symptomatic Contacts

The demographic and clinical characteristics of contacts with TB symptoms (*n* = 240) who underwent further evaluation by sputum tests (*n* = 169) versus those who did not undergo further evaluation by sputum tests (*n* = 71) are presented in Table 1. The age of the contact was the only characteristic that was statistically associated with whether contacts underwent further evaluation by sputum examination or not. Children aged less than 15 years were less likely to have had a sputum examination (either smear for AFB or Xpert test MTB/RIF), and contacts older than 49 years were more likely to have had a sputum examination when compared to contacts in the 15–49 year age group. 

### 3.2. Characteristics Associated with Diagnosed with Active TB among Registered Contacts

The demographic and clinical characteristics of 610 household contacts screened for active TB and the yield/prevalence of active TB in association with these characteristics are shown in Table 2. Overall 58% of the contacts were female, the median age of all contacts (IQR) was 31 (16–46) years and 40 (7%) contacts were children aged less than 5 years. Seventeen (3%) contacts had a previous history of TB, 6 (1%) had positive HIV status, and <1% of the contacts had a history of diabetes mellitus. Children aged less than 5 years had a significantly higher yield of TB when compared to contacts in the adult age groups. Since a small number of contacts were diagnosed with TB (*n* = 24), we did not perform a multivariable analysis to calculate the adjusted prevalence ratios. 

## 4. Discussion

This is the first study describing and evaluating the process of the systematic screening and investigation of household contacts of index MDR-TB patients in Myanmar. The study identified major gaps in the implementation of the screening as per the contact investigation algorithm. About 20% of all contacts enrolled were not screened by chest radiography. A third of the contacts with TB symptoms were not investigated by any sputum examination, and only one-fifth of contacts with TB symptoms were investigated by both sputum smear microscopy and the Xpert MTB/RIF assay. About 4% of contacts were diagnosed as having active TB disease. Children under 5 years of age who were contacts were more likely to be diagnosed with active TB. Since the study used routinely collected project data, we strongly believe that the findings can inform the national program in scaling up MDR-TB household contact screening in Myanmar. 

There are a few limitations to the study. First, we did not have information on the total number of contacts of 210 index cases (the denominator) of which 620 were enrolled. This was due to a gap in our recording system that may have led to the focal nurses enrolling only those who they were able to meet and perform the symptom screening. Therefore, the gap between the number of contacts eligible and the number screened is likely to be higher than shown in our study. Second, as this study was cross-sectional in design, the results only provide an estimate of the prevalence of TB cases among contacts at a certain time period. Since the household contacts are more likely to develop TB anytime following exposure to the index case, a longitudinal study that provides information on both prevalent and incident cases would have provided much better estimates of the actual yield of TB among contacts. Third, due to the cross-sectional nature of the study and also since genotyping of contact’s mycobacterial specimens was not done, we are unable to assess the temporal relationship between the exposure to the index patients and development of TB in the contacts, and therefore we cannot make any inferences about whether the TB disease diagnosed among contacts is due to TB transmission within the households. Fourth, about one-third of contacts did not undergo diagnostic evaluation according to the screening algorithm, and therefore the yield of MDR-TB cases among household contacts of MDR-TB patients in our study is an underestimate of the true yield. Finally, the study was based on routinely collected program data, and therefore there could be some errors in recording and reporting. We did not estimate the magnitude of these errors. However, we believe that due to the supervision and monitoring protocols in place, these errors are likely to be minimal and random, and therefore these errors are unlikely to have a major influence on the study results.

Despite these limitations, the study has some key findings to inform the program and future research. About 80% of contacts as per the screening algorithm underwent chest radiography. Anecdotally we were informed by the field-level health workers that this required substantial resources, time, and effort from them as well as the household contacts. Therefore, the large proportion of contacts who underwent chest radiography in our study may not be sustainable or replicable in routine practice. Therefore, whether chest radiography is required for all household contacts irrespective of the presence of symptoms is a subject matter for further exploration and future study. In this future study, we suggest that different screening and investigation algorithms are compared and tested for their efficacy and cost-effectiveness in detecting active TB among household contacts of MDR-TB patients [11]. 

One-third of household contacts with TB symptoms did not undergo further sputum evaluation. A study from South Africa also showed that less than half of the symptomatic household contacts of MDR-TB underwent further TB diagnosis evaluation [12]. Similarly, many other studies have reported high drop-out rates during TB contact investigation [13]. There are possible patient-level and health system-level barriers that prevent the systematic contact investigation algorithm from being followed. A study conducted in Vietnam reported that contacts and patients’ knowledge, attitude, and practices regarding TB influenced continued engagement in the TB investigation process [14]. Another study from Uganda reported that stigma about TB, the constraint on time and space in clinics for counselling, mistrust of health-center staff by patients and contacts, and high travel costs for health staff to conduct contact screening and for contacts to travel to health facilities were barriers to implement TB contact screening [15].

In addition, not all contacts who were tested by sputum smear examination were tested by Xpert MTB/RIF. Studies have shown that Xpert MTB/RIF can detect up to 59% additional TB cases when compared to sputum smear microscopy [16,17,18]. It can also detect RR-TB as well as reduce the turnaround time from sample collection to diagnosis and treatment [19]. Inadequate access or lack of access to Xpert MTB/RIF machines was one of the main barriers for Xpert MTB/RIF testing for all eligible patients. During the study period, there were only 65 Xpert MTB/RIF machines in the country while there were 330 townships with an MDR-TB center. Many townships did not have Xpert MTB/RIF machines, and some of the townships were far from those that had a functioning machine for referral. The national program has a plan to increase the number of machines in the country (85 machines by the end of 2018), and the NTP’s drug-resistant TB guidelines (February 2017) also recommend screening of household contacts of MDR-TB using Xpert MTB/RIF [9]. This could substantially reduce the barriers for Xpert MTB/RIF testing and increase the number of TB cases detected among the household contacts. Other patient- and provider-level barriers for accessing Xpert MTB/RIF should be explored in this context. 

The prevalence of TB among household contacts in our study is similar to other studies conducted in high TB and MDR-TB burden countries [4,20,21,22]. However, we believe that due to several gaps in implementation and the limitations mentioned above, the prevalence in our setting is likely to be higher than what we observed in this study. In order to obtain more accurate estimates of the burden of TB among household contacts, the gaps and limitations identified in our study must be addressed. This includes close and active surveillance for 24 months for early detection of active disease in those who may be infected [23]. In addition, there is a need to support index patients to improve infection control measures at the household level, such as simple measures to improve cross-ventilation so that further transmission can be minimized [20,23,24]. 

We found that child contacts younger than five years had the highest risk of being diagnosed with TB. Although some studies and systematic reviews have reported that the yield among children is comparable to that seen in adult contacts [5,25], some studies have shown that there is a high prevalence of TB in children among contacts, as seen in our study [26,27]. This can be explained by the fact that young children are more likely to stay at home, which can increase exposure time especially if the index cases are their first-degree relatives [28]. Hence, it would be worthwhile to consider chemoprophylaxis in children after active TB is excluded to prevent the development of TB or MDR-TB [29,30,31]. Currently, we do not have national guidelines on how to manage TB infection in child contacts of MDR-TB, and there is no consensus on the preventive regimen for contacts of MDR-TB. Therefore, there is a need to develop guidelines to manage childhood contacts of patients with MDR-TB and to provide preventive therapy in Myanmar. In the meantime, as per the existing strategy, the program should maintain active surveillance of all contacts so as to detect and treat cases early [32].

### Policy and Practice Implications

The project needs to (1) strengthen the listing of all household contacts in the contact register and continue to record the results of the screening process in a systematic manner; (2) evaluate the efficacy and effectiveness of different contact screening algorithms and identify the most cost-effective and convenient algorithm that can be used in this setting; and (3) identify and address individual and system-level barriers for sputum smear examination and Xpert MTB/RIF testing.

## 5. Conclusions

The yield of TB (~4%) from screening household contacts of index MDR-TB patients was similar to what has been reported from other parts of the world. However, there were major gaps in screening according to the algorithm, and sputum smear microscopy and Xpert MTB/RIF testing were not done in all of the eligible contacts. The project should strengthen the systematic screening and investigation of TB in household contacts of MDR-TB patients, and the NTP should scale up the contact screening for all MDR-TB patients countrywide in order to achieve early detection and treatment of TB and MDR-TB. 

## Figures and Tables

**Figure 1 tropicalmed-05-00002-f001:**
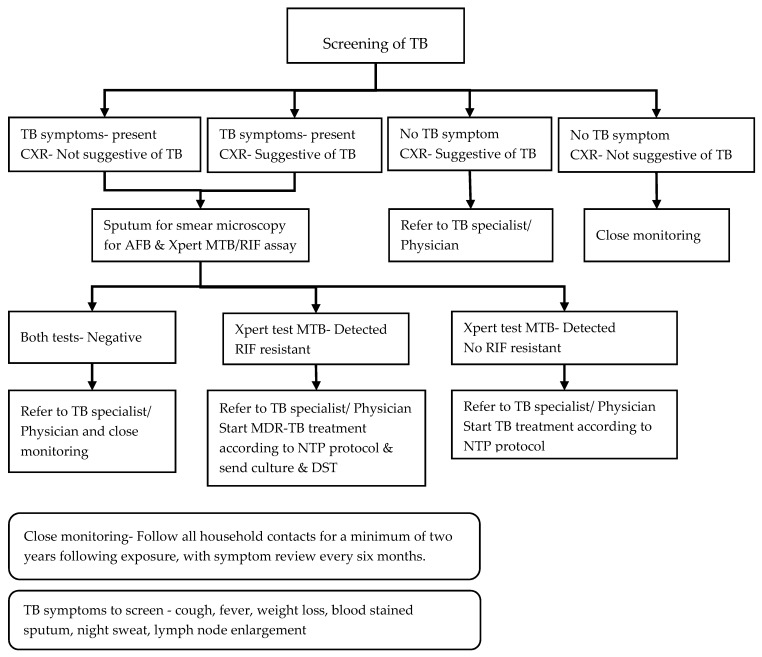
Systematic screening and investigation algorithm for household contacts of index multidrug-resistant tuberculosis (MDR-TB) patients in the community-based MDR-TB care project in Myanmar. TB = tuberculosis; CXR = chest radiography; AFB = acid-fast bacilli; NTP = National TB Programme; DST = drug sensitivity testing.

**Figure 2 tropicalmed-05-00002-f002:**
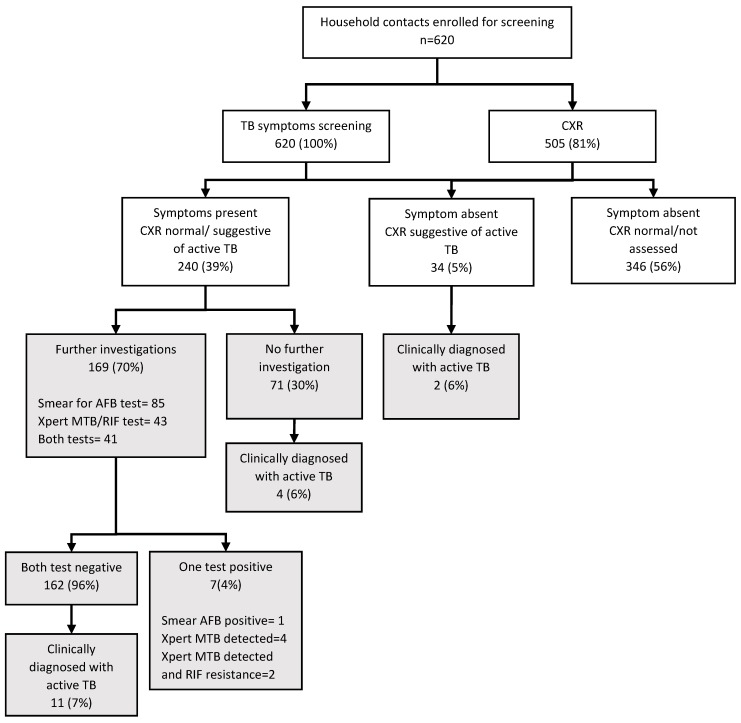
Number of household contacts of MDR-TB patients who underwent TB screening and investigations under the community-based MDR-TB Care Project in 33 townships in Myanmar, April 2016–March 2017. TB = tuberculosis; CXR = chest radiography; AFB = acid-fast bacilli.

**Table 1 tropicalmed-05-00002-t001:** Characteristics of symptomatic household contacts of MDR-TB patients, and their association with following the systematic screening algorithm under the community-based MDR-TB Care Project in 33 townships in Myanmar, April 2016–March 2017.

Characteristics	Total		Followed the Systematic Screening Algorithm			
	*n*	(%) ^†^	*n*	(%) ^§^	OR	(95% CI) *
Total	240	(100)	169	(70.4)		
Sex						
Male	105	(43.7)	70	(66.7)	Ref	
Female	135	(56.3)	99	(73.3)	0.7	(0.3–1.5)
Age						
<5 years	16	(6.7)	2	(12.5)	0.3	(0.0–0.2)
5–14 years	44	(18.3)	15	(34.1)	0.2	(0.0–0.4)
15–49 years	117	(48.8)	95	(81.2)	Ref	
>49 years	58	(24.2)	56	(96.5)	6.5	(2.2–18.9)
Missing	5	(2.1)	1	(20.0)	0.1	(0.0–0.8)
History of previous TB						
Yes	11	(4.6)	9	(81.8)	1.9	(0.5–6.6)
No	229	(95.4)	160	(69.9)	Ref	
HIV status						
Positive	3	(1.3)	3	(1.8)	NA	
Unknown	237	(98.7)	166	(98.2)		
History of diabetes mellitus						
Yes	2	(0.8)	1	(50.0)	0.4	(0.0–6.8)
Unknown	238	(99.2)	168	(70.6)	Ref	

OR = odds ratio; CI = confidence interval; Ref = reference group; * CIs are adjusted for clustering at household level as well as township level. ^†^ Column percentage; ^§^ Row percentage of total contact number. NA = not applicable.

**Table 2 tropicalmed-05-00002-t002:** Demographic and clinical characteristics of household contacts of MDR-TB patients and the yield of TB among household contacts under the community-based MDR-TB Care Project in 33 townships in Myanmar, April 2016–March 2017.

Characteristics	Total		Diagnosed with Active TB				
	*n*	(%) ^†^	*n*	%	(95% CI) ^§,^*	PR	(95% CI) *
Total	610	100	24	3.9	(2.3–6.5)	-	
Sex							
Male	258	(41.6)	12	4.7	(1.9–10.8)	1.4	(0.6–3.4)
Female	362	(58.4)	12	3.3	(2.1–5.2)	ref	
Age							
<5 years	40	(6.5)	4	10.0	(3.6–24.7)	3.7	(1.2–11.4)
5–14 years	98	(15.8)	4	4.1	(1.2–13.4)	1.5	(0.4–5.5)
15–49 years	337	(54.4)	8	2.7	(1.4–5.1)	ref	
>49 years	137	(22.1)	5	4.4	(1.8–10.0)	1.6	(0.6–4.6)
Missing	8	(1.3)	1	12.5	(1.3–60.7)	4.7	(0.6–34.9)
History of previous TB							
Yes	17	(2.7)	0	-			
No	603	(97.3)	24	4.0	(2.3–6.7)	NA	
HIV status							
Positive	6	(1.0)	1	16.7	(1.8–67.9)	4.4	(0.6–32.0)
Unknown	614	(99.0)	23	3.7	(2.2–6.4)	ref	
History of diabetes mellitus							
Yes	3	(0.5)	0	-			
Unknown	617	(99.5)	24	3.9	(2.3–6.6)	NA	

PR = prevalence ratio; CI = confidence interval; Ref = reference group; * CIs are adjusted for clustering at household level as well as township level. ^†^ Column percentage. ^§^ Row percentage of total contact number. NA = not applicable as PR cannot be calculated as there is zero prevalence in one of the two groups.

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
