# Peer review of "Outcomes of Community-Based Systematic Screening of Household Contacts of Patients with Multidrug-Resistant Tuberculosis in Myanmar"

_tropicalmed, 2019, doi:10.3390/tropicalmed5010002_

Round 1

Reviewer 1 Report

This is a well-written analysis of household contact tracing in Myanmar. A total of 24 cases has been detected (almost 4% of contacts), calling for adequate actions within the health sector. 

I have only some minor comments:

Line 31: Please include in abstract the absolute number and additional information: 24 contacts were diagnosed with active TB, of these two with rifampicin resistance.

Line 75: Change “This was a cross-sectional descriptive study of routinely collected program data. “ into “This was an analysis of routinely collected program data.”

Line 182. ”There were 240 (39%) contacts had one or more TB symptoms”: include “that” after “contacts”

Author Response

Dear Editor

Tropical Medicine and Infectious Disease Journal

Thank you and the two reviewers for taking time to review our paper. We have attempted to address all the comments from the reviewers and provide below a point-by-point response and the changes made in the revised manuscript. We submit two copies of our revised manuscript – a version with track changes and a clean version. The line number in the response corresponds to the line number in the track change version.

We hope the paper may now be suitable for publication in the journal. I would like to take this opportunity to say thank you for the very short turn-around time of the review process.

Yours Sincerely

Nang Thu Thu Kyaw (on behalf of my co-authors)

Response to Reviewer 1 Comments

Point 1: This is a well-written analysis of household contact tracing in Myanmar. A total of 24 cases has been detected (almost 4% of contacts), calling for adequate actions within the health sector.

I have only some minor comments:

Line 31: Please include in abstract the absolute number and additional information: 24 contacts were diagnosed with active TB, of these two with rifampicin resistance.

Response 1: We included the additional information in the revised abstract and the sentence reads “Twenty-four contacts were diagnosed with active TB including two rifampicin resistant cases [yield of active TB = 3.9% (95% CI: 2.3-6.5%)]. (Line: 31-32)

Point 2: Line 75: Change “This was a cross-sectional descriptive study of routinely collected program data. “ into “This was an analysis of routinely collected program data.”

Response 2: We have revised the sentence as “This was an analysis of routinely collected program data.”. (Line: 76)

Point 3: Line 182. ”There were 240 (39%) contacts had one or more TB symptoms”: include “that” after “contacts”

Response 3: We have changed to “There were 240 (39%) contacts who had one or more TB symptoms..” in the revised manuscript. (Line: 187)

Reviewer 2 Report

important work. well written, generally . .

7 of 24 diagnoses among contacts based on sputum; was sputum collection standardised in training of focal nurses? spot specimens or morning?

sputum collection was best among contacts >49 yrs - were they smokers?

minor points: line 182 change into ' there were 240 (39%) contacts that were (add 'that')

line 184 the remainder 71: the remaining 71, or: the remainder - 71 (noun, not adjective)

lines 188-9 number to diagnose was 26; is there a confidence interval to this pint estimate?

line 236 .. were not be screened . . delete ' be' 

Author Response

Dear Editor

Tropical Medicine and Infectious Disease Journal

Thank you and the two reviewers for taking time to review our paper. We have attempted to address all the comments from the reviewers and provide below a point-by-point response and the changes made in the revised manuscript. We submit two copies of our revised manuscript – a version with track changes and a clean version. The line number in the response corresponds to the line number in the track change version.

We hope the paper may now be suitable for publication in the journal. I would like to take this opportunity to say thank you for the very short turn-around time of the review process.

Yours Sincerely

Nang Thu Thu Kyaw (on behalf of my co-authors)

Response to Reviewer 2 Comments

Point 1: Moderate English changes required

Response 1: Thank you for your suggestion and we have corrected English in the revised manuscript.

Point 2: The authors addressed an important aspect to fight MDRTB in low resource settings.

The paper reads well; only few typo’s detected.

A chain is as strong as its weakest link - diagnosing TB by sputum, and even more so, diagnosing MDRTB using Xpert strongly depends on adequate sputum samples.

Lines 109-110:  were these sputum samples spot-specimens, or spot-early morning, or --  how exactly were these focal nurses instructed to make sure that adequate sputum samples were obtained?  Instructions to get good sputum specimens is critically important; perhaps elderly smokers can easily produce spot sputum specimens, and many other can’t - unless properly instructed.

Response 2: We agree that adequate and high-quality sputum samples are crucial for TB diagnosis. In this study, an early morning sputum sample was used for Xpert MTB/RIF test. One spot and one early morning sputum sample were used for smear microscopy. The focal nurses are trained to instruct contacts on how to expectorate sputum according to the guidelines. The instruction includes first, rinse the mouth with clean water, second, take a deep breath in and out three times, third, take one deep breath and cough forcefully and finally, spit the sputum into the sputum container provided.

We have added this additional information in the revised manuscript. (Line: 110-115)

Point 3: Figure 2 shows that half of sputum samples were only submitted for AFB. Low adherence? Technical problems to get the Xpert test? Please comment.

Response 3: We did not assess the reasons for not testing sputum for Xpert MTB/RIF. We suspected that lack of access was the main issue because Xpert machines were not installed in every township and some townships were far from the townships where the X-pert MTB/RIF assay was available. We discussed this in our discussion section as “Inadequate access or lack of access to Xpert MTB/RIF machines: During the study period, there were only 65 Xpert MTB/RIF machines in the country while there were 330 townships with an MDR-TB center. Many townships did not have an Xpert MTB/RIF machine and some of the townships were far from the those that had a functional machine for referral” (Line: 291-296)

Point 4: This might in part explain why perhaps in lines 198-199, the age group >49 yr was able to provide sputum specimens; is smoking common among males/females? Specific age groups that smoke? please comment.

Response 4: We agree that the older age group was more likely to undergo sputum testing. However, we did not collect the data on smoking status among contacts and we are not in a position to comment on this.

Point 5: Line 182 There were 240 (39%) contacts had one or more TB symptoms .. CHANGE: There were 240 (39%) contacts that had one or more TB symptoms . .

Response 5: We have changed to “There were 240 (39%) contacts who had one or more TB symptoms..” in the revised manuscript. (Line: 187)

Point 6: Line 184  The remainder 71 (30%) contacts . . CHANGE: The remaining 71 (30%) contacts . . OR :  The remainder  - 71 (30%) contacts . . (grammar issue: remainder is a noun, not an adjective)

Response 6: We have changed to “The remainder - 71 (30%) contacts …” in the revised manuscript. (Line: 189)

Point 7: Lines 188-189: The number of household contacts screened to diagnose one case of active TB was 26. QUESTION:  would it be possible to provide a confidence interval here, or any other measure to give some insight in the precision?

Response 7: We have added the confidence interval in the revised manuscript. (Line 194)

Point 8: Line 236 contacts enrolled were not be screened .. CHANGE:  . . were not screened

Response 8: We have corrected this in the revised manuscript. (Line 241)